# Dual BACE1 and Cholinesterase Inhibitory Effects of Phlorotannins from *Ecklonia cava*—An In Vitro and in Silico Study

**DOI:** 10.3390/md17020091

**Published:** 2019-02-01

**Authors:** Jinhyuk Lee, Mira Jun

**Affiliations:** 1Korean Bioinformation Center, Korea Research Institute of Bioscience and Biotechnology (KRIBB), 125, Gwahak-ro, Yuseong-gu, Daejeon 34141, Korea; jinhyuk@kribb.re.kr; 2Department of Bioinformatics, KIRBB School of Bioscience, Korea University of Sciences and Technology, 217 Gajung-ro, Yuseong-gu, Daejeon 34113, Korea; 3Department of Food Science and Nutrition, College of Health Sciences, Dong-A University, 37, Nakdong-daero 550 beon-gil, Saha-gu, Busan 49315, Korea; 4Center for Silver-Targeted Biomaterials, Brain Busan 21 Plus Program, Graduate School, Dong-A University, Nakdong-daero 550 beon-gil, Saha-gu, Busan 49315, Korea; 5Institute of Convergence Bio-Health, Dong-A University, Busan 49315, Korea

**Keywords:** Alzheimer′s disease, BACE1, acetylcholinesterase, in silico docking, phlorotannins

## Abstract

Alzheimer′s disease (AD) is one of the most common neurodegenerative diseases with a multifactorial nature. β-Secretase (BACE1) and acetylcholinesterase (AChE), which are required for the production of neurotoxic β-amyloid (Aβ) and the promotion of Aβ fibril formation, respectively, are considered as prime therapeutic targets for AD. In our efforts towards the development of potent multi-target, directed agents for AD treatment, major phlorotannins such as eckol, dieckol, and 8,8′-bieckol from *Ecklonia cava* (*E. cava*) were evaluated. Based on the in vitro study, all tested compounds showed potent inhibitory effects on BACE1 and AChE. In particular, 8,8′-bieckol demonstrated the best inhibitory effect against BACE1 and AChE, with IC_50_ values of 1.62 ± 0.14 and 4.59 ± 0.32 µM, respectively. Overall, kinetic studies demonstrated that all the tested compounds acted as dual BACE1 and AChE inhibitors in a non-competitive or competitive fashion, respectively. In silico docking analysis exhibited that the lowest binding energies of all compounds were negative, and specifically different residues of each target enzyme interacted with hydroxyl groups of phlorotannins. The present study suggested that major phlorotannins derived from *E. cava* possess significant potential as drug candidates for therapeutic agents against AD.

## 1. Introduction

Alzheimer′s disease (AD) is a progressive and irreversible neurodegenerative disorder with characteristic features of cognitive dysfunction, memory impairment, and behavior disturbances. The neuropathological hallmarks of AD patients are the presence of extracellular deposits of amyloid plaques and intracellular filamentous neurofibrillary tangles in the brain [1]. Amyloid plaques and neurofibrillary tangles are aggregates of amyloid-β peptide (Aβ) and hyperphosphorylated tau protein, respectively. In recent years, the “amyloid hypothesis” has arisen as the major pathological mechanism in AD, and the evidence from transgenic mice models revealed that Aβ triggered tau phosphorylation and neurofibrillary tangles formation [2].

Aβ is generated by the sequential proteolytic cleavage of two aspartic proteases, β- and γ–secretase, in the amyloidogenic pathway. β-Secretase (BACE1) initially cleaves amyloid precursor protein (APP) at the N-terminus of the Aβ peptide domain, which is followed by the cleavage of γ–secretase in the transmembrane region of APP, leading to the production of Aβ peptide [3]. Therefore, these secretases have been suggested as potential targets to hinder Aβ formation and thereby delay or stop the progression of AD. γ-Secretase inhibitors have shown severe toxicity problems because of on-target interference with Notch signaling, and genetic deletion of catalytically active subunit presinilin-1 (PS-1) was found to be lethal in embryonic mice [4,5,6]. In contrast, BACE1 inhibition powerfully lowers the level of Aβ in the central nervous system (CNS) of both transgenic mice models and AD patients [7,8,9]. Recent studies have exhibited few mechanism-based side effects of BACE1 inhibition with chronic administration in animal models, but these are relatively weak and mild compared to γ-secretase-inhibitor-induced deficits [10,11,12]. Unlike BACE1, which leads to the formation of Aβ, α-secretase acts within the Aβ domain to preclude the Aβ generation in the non-amyloidogenic pathway. The α-secretase and BACE1 compete for the same APP substrate, with an increase in one cleavage event leading to a decrease in the other. As BACE1 initiates the amyloidogenic pathway and is putatively rate-limiting, it is a critical target for lowering cerebral Aβ levels in the treatment and/or prevention of AD.

In the past, it has been found that acetylcholinesterase (AChE) is involved in the degradation of the neurotransmitter. The observation of a significant loss of cholinergic neurons in AD patients is the major correlate of cognitive impairment. Cholinesterase inhibitors can increase acetylcholine (ACh) levels in the synaptic cleft and partially ameliorate cognitive symptoms for patients with mild to severe AD [13]. Recent findings supported that AChE is associated predominantly with pre-amyloid diffuse deposits, amyloid cores of mature amyloid plaques, and cerebral blood vessels in an AD patient brain. In addition, it triggers the Aβ fibrillogenesis via the formation of stable Aβ– AChE complexes [14,15]. Neurons treated with these complexes exhibited a disrupted neurite network compared to neurons treated with Aβ alone [16]. Based on these findings, the suppression of both enzymes is a very desirable feature of AD therapy.

Current AD therapies are mainly palliative and temporarily slow cognitive decline, and treatments based on the underlying pathologic mechanisms of AD are totally limited [17]. Several therapeutic approaches have recently revealed promising results in clinical trials, such as BACE1 and γ-secretase inhibitors, inhibition of Aβ plaque formation, passive Aβ immunotherapy, etc. However, the clinical use of these agents needs further careful assessment of their effectiveness on cognitive decline and their adverse effects [18]. Another strategy for AD therapy is the use of natural products, which are more effective, safer and have fewer adverse effects than synthesized drugs [19]. Neuroprotective natural compounds such as (-)-epigallocatechin-3-gallate (EGCG) from green tea, resveratrol from grape, curcumin from tumeric, and quercetin from apples revealed significant therapeutic potential toward the amelioration and prevention of AD [20].

Marine organisms are a rich source of several natural molecules, including polyphenol, polysaccharide, sterol, and peptide, which have many biological properties such as antioxidant, anti-inflammatory, anti-hypertensive, anti-obesity, anti-diabetes, and anti-cancer effects [21,22,23,24,25]. *Ecklonia cava* (*E. cava*) is an edible brown seaweed which is distributed in Japan and the southern coast of Korea, and it is recognized as a rich source of bioactive derivatives, containing 3.1% crude phlorotannins [26,27]. Phlorotannins are unique polyphenolic compounds containing a dibenzo[1,4]dioxin element as the core structure not found in terrestrial plants. The compounds consist of phloroglucinol units linked to each other in several ways. Based on the type of linkage, phlorotannins are classified into four subgroups: eckols (phlorotannins with a dibenzodioxin linkage), fuhalols and phlorethols (with an ether linkage), fucols (with a phenyl linkage), and fucophloroethols (with an ether and phenyl linkage) [27]. 

Recently, it has been reported that phlorotannins possess various bioactivities such as antioxidant, antidiabetic, anti-hypertensive, anti-human-immunodeficiency-virus type-1 (HIV-1), and radioprotective activities [28,29,30,31,32]. Regarding the study of neuroprotective effects, phlorotannins-rich *E. cava* extract ameliorated the Aβ formation by modulating α- and γ-secretase expression and inhibiting Aβ-induced neurotoxicity [19,33]. In our previous study, three major phlorotannins of *E. cava*—eckol, dieckol, and 8,8′-bieckol—exhibited anti-apoptotic and anti-neuroinflammatory properties against Aβ-induced cellular damage, which led to our interest in the study of phlorotannins-mediated suppression of related enzymes in Aβ production and aggregation [34]. Therefore, the purpose of the present study is to evaluate the inhibitory effects of these compounds against both BACE1 and AChE through in vitro and in silico approaches. 

## 2. Results

### 2.1. In Vitro Inhibitory Study of Phlorotannins on BACE1 and AChE

The chemical structures of eckol [4-(3,5-dihydroxyphenoxy)dibenzo-p-dioxin-1,3,6,8-tetrol], dieckol [4-[4-[6-(3,5-dihydroxyphenoxy)-4,7,9-trihydroxydibenzo-p-dioxin-2-yl]oxy-3,5-dihydroxyp henoxy]dibenzo-p-dioxin-1,3,6,8-tetrol], and 8,8′-bieckol [9-(3,5-dihydroxyphenoxy)-2-[9-(3,5-dihydroxyphenoxy)-1,3,6,8-tetrahydroxydibenzo-p-dioxin-2-yl]dibenzo-p-dioxin-1,3,6,8-tetrol] were shown in Figure 1. Eckol is a trimer of phloroglucinol (1,3,5-trihydroxybenzene) units, while dieckol and 8,8′-bieckol are hexamer. As presented in Table 1 and Figure 2, 8,8′-bieckol exhibited the strongest BACE1 inhibition (IC_50_, 1.62 ± 0.14 µM), followed by dieckol (IC_50_, 2.34 ± 0.10 µM) and eckol (IC_50_, 7.67 ± 0.71 µM). Interestingly, the IC_50_ value of all of the tested compounds were lower than that of resveratrol (IC_50_, 14.89 ± 0.54 µM), which is a well-known BACE1 inhibitor that was used as a positive control.

Three phlorotannins displayed high potencies as AChE inhibitors (Table 1 and Figure 2). Both dieckol and 8,8′-bieckol exhibited potent AChE inhibitory activity (IC_50_ values of 5.69 ± 0.42 and 4.59 ± 0.32 µM, respectively) and about twofold greater than the eckol (IC_50_, 10.03 ± 0.94 µM).

To demonstrate the specificity of the targeted enzymes, all compounds were tested against tumor necrosis-converting enzyme (TACE), which is a candidate for α-secretase, and other serine proteases, including trypsin, chymotrypsin, and elastase (Table 2). With serine proteases being found in nearly all body tissues and involved in various physiological functions, including digestion, reproduction and immune response, off-target activity causing severe side effects is possible and likely if an inhibitor is not specific to BACE1 and AChE [35]. Therefore, to determine whether phlorotannins inhibited only targeted enzymes without affecting the normal pathway, all compounds were tested against TACE and serine proteases. Up to 100 μM, our tested compounds did not show significant inhibition against the above enzymes, indicating that three phlorotannins appeared to be relatively specific inhibitors of BACE1 and AChE.

### 2.2. Kinetic Study of BACE1 and AChE Inhibition

As shown in Table 1 and Figure 3, Lineweaver–Burk plots for the inhibition of BACE1 by eckol, dieckol, and 8,8′-bieckol were fitted well to the noncompetitive inhibition mode in visual inspection, and the *K*_i_ values of eckol, dieckol, and 8,8′-bieckol were 31.2, 20.1, and 13.9 µM, respectively. On the other hand, our tested compounds were competitive inhibitors of AChE, where the Lineweaver– Burk plots intersected a common point on the *y*-axis (Table 1 and Figure 4). The *K*_i_ values of eckol, dieckol, and 8,8′-bieckol were 37.3, 12.3, and 11.4 µM, respectively, and were obtained from the Dixon plot.

### 2.3. In Silico Docking Study of the Inhibition of BACE1 and AChE by Phlorotannins

According to the in silico docking simulation results, BACE1 and phlorotannins complexes had an allosteric inhibition mode (Table 3 and Figure 5). GLY34 and SER36 of BACE1 formed two hydrogen bonds with the hydroxyl group of eckol with bonding distances of 3.277 and 3.239 Å, respectively. In the dieckol–BACE1 complex, TRP76, THR232, and LYS321 participated in three hydrogen bonds (bonding distance: 2.960, 3.149, and 3.488 Å, respectively). 8,8′-Bieckol–BACE1 complex had four hydrogen bonding interactions with residues LYS107, GLY230, THR231, and SER325 (bonding distance: 3.120, 2.773, 3.098, and 2.887 Å, respectively). In addition, the lowest binding energy of the three tested compounds were negative values: -8.8 kcal/mol for eckol, -10.1 kcal/mol for dieckol, and -9.0 kcal/mol for 8,8′-bieckol.

As indicated in Table 4 and Figure 6, the docking results for eckol, dieckol, and 8,8′-bieckol indicated negative binding energies of −8.8, −9.5, and −9.2 kcal/mol, respectively. Hydrogen bonding interactions between eckol and THR83, TRP86, TYR124, and SER125 of AChE were observed by five hydrogen bonds (bonding distance of 2.855, 2.712, 3.134, 2.883, and 3.313 Å, respectively). Dieckol was bound at the ASN233, THR238, ARG296, and HIS405 of AChE, linked by four hydrogen bonds with bonding distances of 3.399, 2.837, 3.344, and 3.181 Å, respectively, while 8,8′-bieckol had one hydrogen bond with the ARG296 residue of AChE (bonding distance: 3.151 Å).

## 3. Discussion

With respect to the development of anti-AD agents, enzyme inhibition is one of the most promising potential therapeutic strategies. Since BACE1 is the initiating and rate-limiting enzyme in Aβ formation, it is considered as a key target for lowering cerebral Aβ levels [7,8,9]. Additionally, AChE plays a critical role in cholinergic neurotransmission and participates in non-cholinergic mechanisms such as accelerating Aβ fibril formation through conformational change of Aβ and increasing Aβ toxicity by Aβ–AChE complexes [14,15,16]. Thus, multi-enzyme target inhibition against BACE1 and AChE may provide a promising therapeutic approach for AD. In search of a candidate for AD prevention and/or treatment, numerous researchers over the past few decades have focused on discovering natural enzyme inhibitors. Several natural inhibitors of BACE1 and AChE such as coumarins, citrus flavanones, triterpenoids, and alkaloids have been reported [36,37,38,39]. However, efforts to explore bioactive constituents from marine organisms against BACE1 and AChE have been relatively limited.

In the current study, three effective phlorotannins—eckol, dieckol, and 8,8′-bieckol—were studied for their inhibitory properties on BACE1 and AChE. These compounds exhibited powerful inhibitory activities on BACE1 with IC_50_ values at a range of 1.6–7.7 µM. Several terrestrial plant-derived BACE1 inhibitors, including hespretin, naringenin, and hesperidin, were from citrus fruits with IC_50_ values ranging from 16.9–30.3 µM. Alkaloids (neferine, liensinine, and vitexin) in *Nelumbo nucifera* (IC_50_ in the 6.4–28.5 µM range) have been proven to be efficient BACE1 inhibitors. Umbelliferone, isoscopoletin, 7-methoxy coumarin, esculetin, and daphnetin from *Angelica decursiva* with IC_50_ ranging from 7.7–172.3 µM were identified as BACE1 inhibitors. Compared with those plant-derived BACE1 inhibitors, our compounds demonstrated predominantly inhibitory properties against BACE1 [36,37,39].

Interestingly, the difference in inhibitory properties among phlorotannins is related to the number of hydroxyl groups present. In our new findings, it was shown that 8,8′-bieckol containing 11 OH groups had the highest inhibitory efficacy against BACE1 when compared to dieckol (10 OH groups) and eckol (6 OH groups). When phlorotannins from *Eisenia bicyclis*, one of the brown algae, were investigated for their BACE1 inhibitory effects, the result that dieckol was stronger than eckol was similar to that of our present study [40]. Consistent with our result, Ahn and colleagues have reported that the inhibitory effect on HIV-reverse transcriptase of 8,8′-bieckol containing a biaryl linkage was tenfold higher than that of 8,4′′′-dieckol with a diphenyl ether linkage [41]. This observation indicated that the steric hindrance of the hydroxyl and aryl groups near the biaryl linkage of 8,8′-bieckol noticeably enhanced its inhibitory potency. 

Among three phlorotannins, 8,8′-bieckol showed the most potent inhibitory activity against AChE. Similar results regarding the correlation between the molecular size of phlorotannins and enzyme inhibitory efficacy was revealed in a previous study. 8,8′-bieckol showed more potent activity against hyaluronidase, with an IC_50_ of 40 µM, than dieckol and eckol (IC_50_, 120 and >800 µM, respectively) [42]. In addition, the present study first demonstrated the specific molecular docking interaction as well as biological properties of eckol, dieckol, and 8,8′-bieckol against AChE.

The BACE1 inhibition kinetics indicated that the tested compounds act as non-competitive inhibitors, which means that these compounds can bind either another regulatory site or to the subsite of BACE1. The inhibition level is dependent on the concentration of the inhibitor but is not reduced by increasing concentrations of substrate. Because of this, V_max_ is reduced, but K_m_ is unaffected. In BACE1 inhibitory activity, phlorotannins decreased the V_max_ values without affecting the affinity of BACE1 toward the K_m_, which demonstrated that these compounds exhibited non-competitive inhibition against BACE1. However, AChE kinetics results exhibited that our tested compounds are competitive inhibitors with unchanged V_max_ and increased K_m_. In other words, these compounds interacted directly with the catalytic site of AChE instead of with other allosteric pockets.

In silico docking analysis is a valuable drug discovery tool and can be used to discover prospective, biologically active molecules from natural product databases. The results of the molecular docking score were provided to evaluate the capacity of different protein–ligand complex interactions and to compare the biological activities and the inhibition mode. In the BACE1 docking simulation, multiple hydrogen interactions were observed in the BACE1–phlorotannins complexes. Eckol interacted with both GLY34 and SER36 of BACE1, and dieckol bounded to TRP76, THR232, and LYS321. In addition, 8,8′-bieckol formed four hydrogen bonds with BACE1 residues, including LYS107, GLY230, THR231, and SER325. These docking results showed that hydrogen bonds between phlorotannins and allosteric residues of BACE1 play an important role in enzyme inhibition. 

AChE docking analysis provides insight into the mechanism underlying active site binding interaction. The hydroxyl group of eckol formed five hydrogen bonds with THR83, TRP86, TYR124, and SER125 of AChE. In particular, the choline-binding site residue (TRP86) of AChE was involved in hydrogen bond interaction with eckol. Dieckol showed four hydrogen-bond interactions with ASN233, THR238, ARG296, and HIS405, whereas 8,8′-bieckol made one hydrogen bond with ARG296 located in the active site of AChE. These docking results from the in silico study were in agreement with our in vitro experimental data.

To date, few studies have investigated the neuroprotective property of phlorotannins. Our previous study demonstrated that phlorotannins ameliorated Aβ_25–35_ toxicity through the regulation of the apoptotic signal and the NF-kB/MAPKs pathway [34]. Eckol and dieckol suppressed H_2_O_2_-induced oxidative stress in murine hippocampus neuronal cells [43]. Moreover, it has been reported in an in vivo study that oral administration of dieckol (10 mg/kg) improved cognitive ability in ethanol-induced memory impairment mice [44]. 

Nagayama and coworkers demonstrated no significant toxic effects in the oral administration of up to 1,500 mg/kg of phlorotannins for 14 days in male and female Institute of Cancer Research (ICR) mice [45]. In a human study, *E. Cava* extract was shown to be safe for use in food supplements at a maximum daily intake level of 263 mg/day for adults [46]. Collectively, phlorotannins are toxicologically very safe, explaining their traditional and present consumption as foods and medicinal products.

Bioavailability parameters such as biotransformation and conjugation during absorption from the GI tract are principle factors influencing in vivo biological activity. Lipinski′s rule of five is a widespread strategy to define bioavailability predictions of drug molecules. According to this predictive model, a compound needs to exhibit optimum GI absorption with a molecular weight of < 500 Da, no more than five hydrogen bond donors, no more than ten hydrogen bond acceptors, and a calculated partition coefficient (LogP) that is no more than five [47]. Fortunately, eckol meets Lipinski′s requirements for acceptable oral bioavailability, while dieckol and 8,8′-bieckol have limitations on bioavailability [48]. However, the compounds absorbed by specific transporters are an exception to this rule, and a recent study demonstrated that dieckol successfully penetrated into the brain via crossing the blood–brain barrier (BBB), suggesting that the compound may be transported through an unknown mechanism [49]. A study of the permeability of eckol and 8,8′-bieckol was limited, but it is likely that similar results might also be predictable as that of dieckol. Overall, our marine compounds from *E. Cava* are safe, potent, and selective natural dual inhibitors against BACE1 and AChE that can be used for the multi-target, directed agents of AD.

## 4. Materials and Methods

### 4.1. General

Fluorescence and optical density were measured by Bio-TEK ELISA fluorescence reader FLx 800 and Bio-TEK ELx 808, respectively (Winooski, VT, USA). Eckol (>95%), dieckol (>95%), and 8,8′-bieckol (>95%), were bought from National Development Institute of Korean Medicine (Gyeongsangbuk-do, Korea). The BACE1 assay kit was purchased from Invitrogen (Pan Vera, Madison, WI, USA). TACE and substrate were bought from R&D Systems (Minneapolis, MN, USA). AChE from *Electrophorus electricus* (electric eel), 5,5′-dithiobis-(2 nitrobenzoic acid) (DTNB), resveratrol, galantamine, trypsin, chymotrypsin, elastase, and their substrates, including N-benzoyl-l-Arg-pNA, N-benzoyl-l-Tyr-pNA, and N-succinyl-Ala-Ala-Ala-p NA, were from Sigma-Aldrich (St. Louis, MO, USA).

### 4.2. Enzyme inhibition Studies

Fluorometric assays with a recombinant human BACE1 or TACE were conducted according to manufacturer instructions. Briefly, reaction mixtures containing human recombinant BACE1 (1.0 U/mL), the substrate (75 μM in 50 mM ammonium bicarbonate), and phlorotannins dissolved in an assay buffer (50 mM sodium acetate, pH 4.5) were incubated in darkness for 60 min at 25 °C in well plates. The increase in fluorescence intensity produced by substrate hydrolysis was observed on a fluorescence microplate reader with excitation and emission wavelengths of 545 and 590 nm, respectively. The inhibition ratio was obtained using the following equation:Inhibition (%) = [1 − (*S* − *S*_0_)/(*C* − *C*_0_)] × 100
where *C* was the fluorescence of control (enzyme, assay buffer, and substrate) after 60 min of incubation, *C*_0_ was the fluorescence of control at time 0, *S* was the fluorescence of tested samples (enzyme, sample solution, and substrate) after 60 min of incubation, and *S*_0_ was the fluorescence of the tested samples at time 0.

A human recombinant TACE (0.1 ppm in 25 mM Tris buffer), the substrate (APP peptide YEVHHQKLV using EDANS/DABCYL), and phlorotannins were dissolved in an assay buffer, which were then combined and incubated for 60 min in the dark at 25 °C. The increase in fluorescence intensity produced by substrate hydrolysis was observed on a fluorescence microplate reader with excitation and emission wavelengths of 320 and 405 nm, respectively.

The colorimetric assays, including AChE, trypsin, chymotrypsin, and elastase were assayed according to previously described methods [36]. The hydrolysis of AChE was monitored according to the formation of yellow 5-thio-2-nitrobenzoate anions at 405 nm for 15 min, which were produced by the reaction of DTNB with thiocholine released from ACh. All reactions were performed in 96-well plates in triplicate and recorded using a microplate spectrophotometer.

*N*-benzoyl-l-Arg-*p*NA, *N*-benzoyl-l-Tyr-*p*NA, and *N*-succinyl-Ala-Ala-Ala-*p*NA were used as substrates to assay the inhibition of trypsin, chymotrypsin, and elastase, respectively. Enzyme, Tris-HCl buffer (0.05 M, in 0.02 M CaCl_2_, pH 8.2), and phlorotannins were incubated for 10 min at 25 °C; then, substrate was added for 30 min at 37 °C. The absorbance was recorded at 410 nm. The inhibition ratio was obtained using the following equation:Inhibition (%) = {[1 − (A − B)]/control} × 100
where A was the absorbance of the control (enzyme, assay buffer, and substrate) after 60 min of incubation, and B was the absorbance of tested sample (assay buffer and sample solution) after 60 min of incubation.

### 4.3. Enzyme Kinetic Study

To evaluate the kinetic mechanisms of phlorotannins towards BACE1 and AChE, Dixon and Lineweaver–Burk plots were conducted by various concentrations of substrate and inhibitors. Kinetic parameters such as *K*_i_, V_max_, and K_m_ values were calculated by Sigma Plot 12.3 (Systat Software, Inc., San Jose, CA, USA)

### 4.4. Molecular Docking Study

X-ray crystal structures of human BACE1 (PDB code: 2WJO) and AChE (PDB code: 4PQE) were retrieved from the Protein Data Bank (PDB, http://www.rcsb.org/). Three-dimensional (3D) structures of eckol, dieckol, and 8,8′-bieckol were obtained from PubChem with compound identification number (CID) of 145937, 3008868, and 3008867, respectively. The Autodock Vina software version 1.1.2 (The Scripps Research Institute, La Jolla, CA, USA,) was used to conduct molecular docking analysis. The dimensions of the grid were 30 × 30 × 30 Å, the cluster radius was 1 Å, and the Cα coordinates in each selected backbone binding residue of the protein receptor was used for the center of docking space. Other options for docking simulations were used as defaults. The atomic coordinates of the ligands were drawn and displayed using Marvin sketch (5.11.4, 2012, ChemAxon, One Broadway Cambridge, MA, USA).

### 4.5. Statistical Analysis

All results were presented as the mean ± SD of three independent experiments. Statistical significance was assessed by Duncan′s multiple range tests using Statistical Analysis System (SAS) version 9.3 (SAS Institute, Cary, NC, USA).

## 5. Conclusions

The integration of enzyme activity, kinetics, and in silico docking studies provided principle insights into the molecular basis underlying ligand binding affinity and BACE1 and AChE inhibition. Accordingly, these results suggested that phlorotannins from *E. cava*, especially 8,8′-bieckol, have noteworthy potential for the possible development as treatments and/or preventative agents against AD.

## Figures and Tables

**Figure 1 marinedrugs-17-00091-f001:**
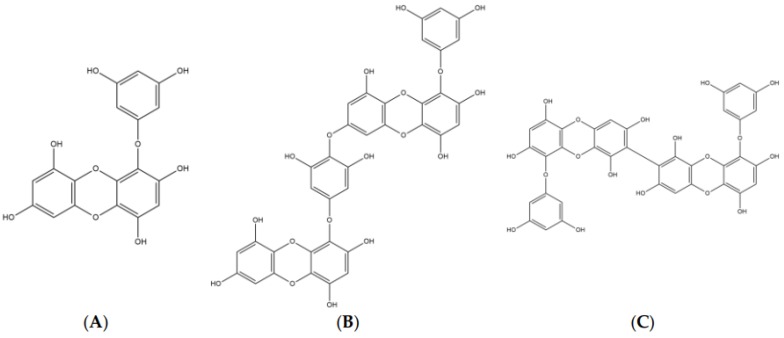
The chemical structures of (**A**) eckol, (**B**) dieckol, and (**C**) 8,8′-bieckol.

**Figure 2 marinedrugs-17-00091-f002:**
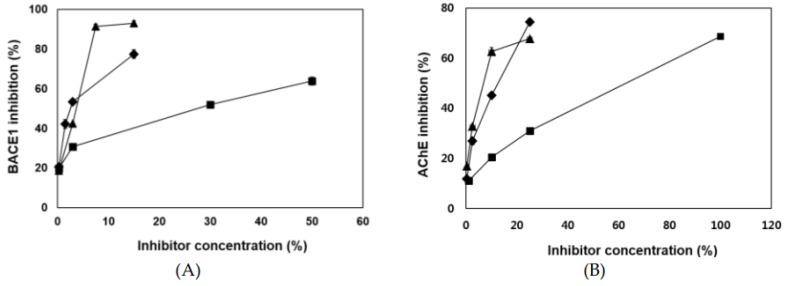
(**A**) β-Secretase (BACE1) and (**B**) acetylcholinesterase (AChE) inhibitory activities of eckol (■), dieckol (◆), and 8,8′-bieckol (**▲**). All assays were performed in three independent experiments. Dimethyl sulfoxide (DMSO) was used as negative controls in the BACE1 and AChE assays.

**Figure 3 marinedrugs-17-00091-f003:**
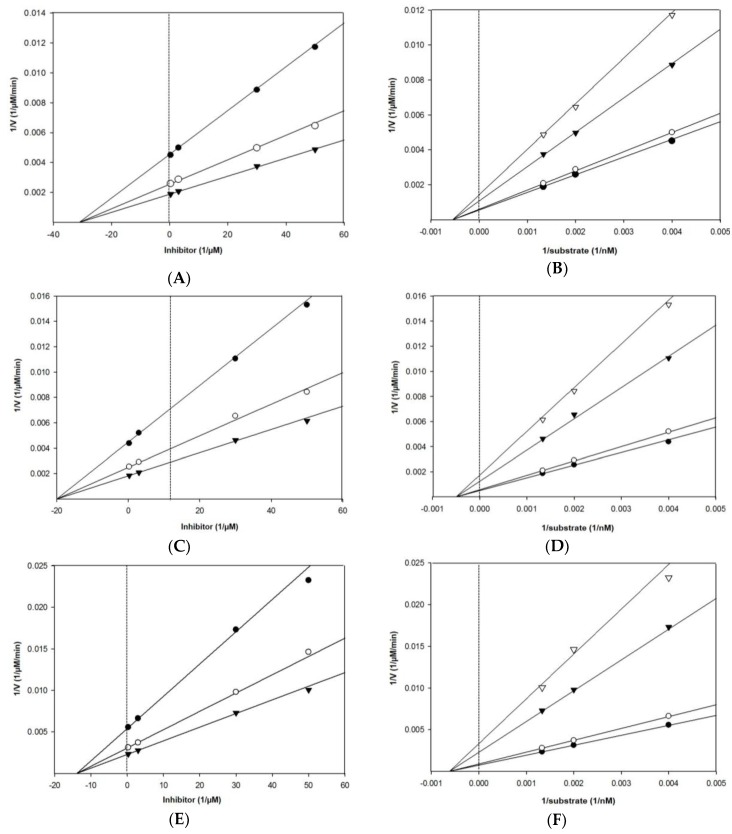
Dixon plot of BACE1 inhibition by (**A**) eckol, (**C**) dieckol, and (**E**) 8,8′-bieckol in the presence of different substrate concentrations: 250 nM (●), 500 nM (○), and 750 nM (▼). Lineweaver–Burk plot of BACE1 inhibition by (**B**) eckol, (**D**) dieckol, and (**F**) 8,8′-bieckol in the presence of different inhibitor concentrations: 0.3 μM (●), 3 μM (○), 30 μM (▼), and 50 μM (▽) for eckol; 0.3 μM (●), 1.5 μM (○), 3 μM (▼), and 15 μM (▽) for dieckol; 0.3 μM (●), 3 μM (○), 7.5 μM (▼), and 15 μM (▽) for 8,8′-bieckol. All assays were performed in three independent experiments.

**Figure 4 marinedrugs-17-00091-f004:**
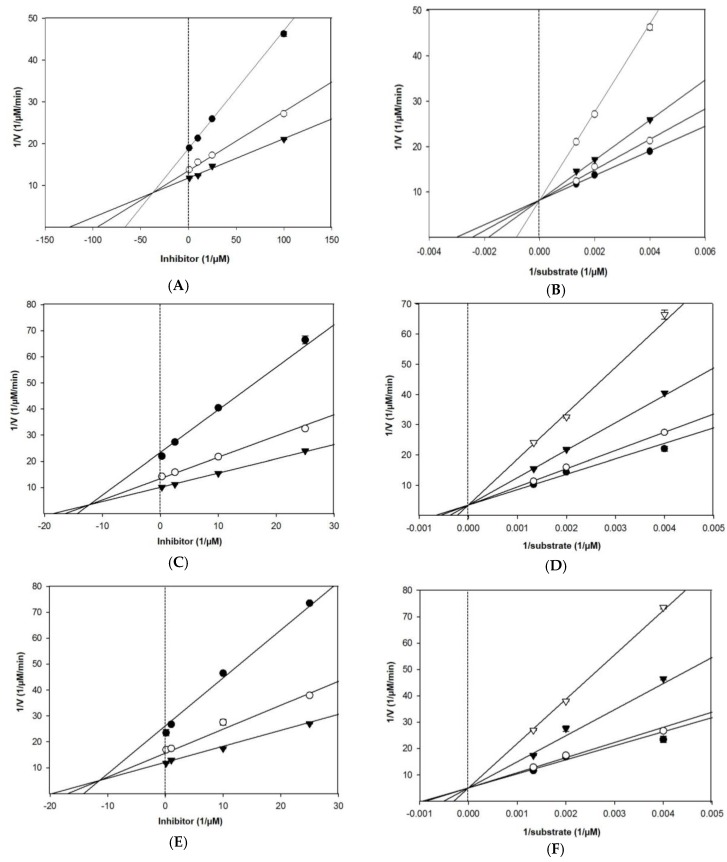
Dixon plot of AChE inhibition by (**A**) eckol, (**C**) dieckol, and (**E**) 8,8′-bieckol in the presence of different substrate concentrations: 250 µM (●), 500 µM (○), and 750 µM (▼). Lineweaver–Burk plot of AChE inhibition by (**B**) eckol, (**D**) dieckol, and (**F**) 8,8′-bieckol in the presence of different inhibitor concentrations: 1 μM (●), 10 μM (○), 25 μM (▼), and 100 μM (▽) for eckol; 0.1 μM (●), 10 μM (○), 25 μM (▼), and 50 μM (▽) for dieckol and 8,8′-bieckol. All assays were performed in three independent experiments.

**Figure 5 marinedrugs-17-00091-f005:**
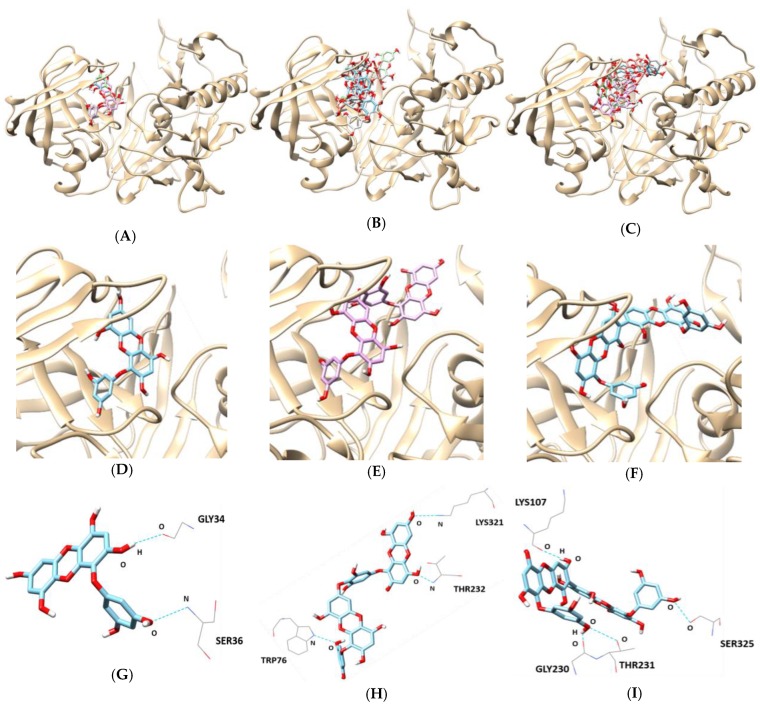
In silico docking simulation of BACE1 inhibition by (**A**) eckol, (**B**) dieckol, and (**C**) 8,8′-bieckol. View of the binding site magnified from (**D**) eckol, (**E**) dieckol, and (**F**) 8,8′-bieckol. Hydrogen interaction diagram of (**G**) eckol, (**H**) dieckol, and (**I**) 8,8′-bieckol.

**Figure 6 marinedrugs-17-00091-f006:**
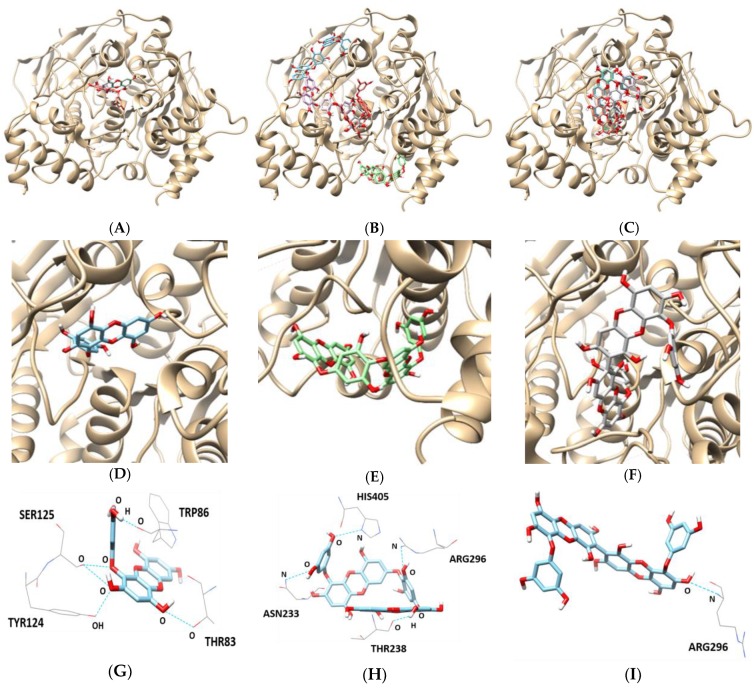
In silico docking simulation of AChE inhibition by (**A**) eckol, (**B**) dieckol, and (**C**) 8,8′-bieckol. View of the binding site magnified from (**D**) eckol, (**E**) dieckol, and (**F**) 8,8′-bieckol. Hydrogen interaction diagram of (**G**) eckol, (**H**) dieckol, and (**I**) 8,8′-bieckol.

**Table 1 marinedrugs-17-00091-t001:** Inhibitory activities of phlorotannins on BACE1 and AChE.

Compounds	IC_50_ (mean ± SD, µM) ^a^		*K*_i_ value (µM) ^d^		Inhibition Mode ^e^	
	BACE1	AChE	BACE1	AChE	BACE1	AChE
Eckol	7.67 ± 0.71	10.03 ± 0.94	31.2	37.3	Non-competitive	Competitive
Dieckol	2.34 ± 0.10	5.69 ± 0.42	20.1	12.3	Non-competitive	Competitive
8,8′-Bieckol	1.62 ± 0.14	4.59 ± 0.32	13.9	11.4	Non-competitive	Competitive
Resveratrol ^b^	14.89 ± 0.54	-	-	-	-	-
Galantamine ^c^	-	0.99 ± 0.07	-	-	-	-

^a^ The IC_50_ values (µM) were calculated from a log dose inhibition curve and expressed as the mean ± standard deviation (SD). All assays were performed in three independent experiments. DMSO was used as a negative control in the BACE1 and AChE assays. ^b^ Resveratrol and ^c^ galantamine were used as positive controls in the BACE1 and AChE assays, respectively. ^d^ Inhibition constant (*K*_i_) and ^e^ inhibition mode were determined using Dixon plot and Lineweaver–Burk plot, respectively.

**Table 2 marinedrugs-17-00091-t002:** Inhibitory activities of phlorotannins against tumor necrosis-converting enzyme (TACE), trypsin, chymotrypsin, and elastase ^a,b^.

Compounds (μM)		TACE (α-Secretase)	Trypsin	Chymotrypsin	Elastase
**Eckol**	50	19.29 ± 1.52	3.59 ± 0.57	1.48 ± 0.19	6.06 ± 0.13
	100	10.60 ± 0.53	4.73 ± 0.25	2.65 ± 0.06	4.37 ± 0.27
Dieckol	50	16.90 ± 1.01	6.44 ± 0.76	4.39 ± 0.44	6.70 ± 0.85
	100	18.33 ± 0.41	−16.64 ± 1.40	1.43 ± 0.02	6.20 ± 0.14
8,8′-Bieckol	50	11.07 ± 0.53	2.98 ± 0.26	0.86 ± 0.02	5.26 ± 0.43
	100	3.57 ± 0.05	−23.05 ± 0.32	0.33 ± 0.01	7.52 ± 0.24

^a^ The inhibitory activity (%) was expressed as the mean ± SD of three independent experiments. DMSO was used as a negative control in TACE and serine proteases assays. ^b^ Comparison of concentration level in each sample is not significantly different.

**Table 3 marinedrugs-17-00091-t003:** Molecular interactions of BACE with eckol, dieckol, and 8,8′-bieckol.

Ligand	Lowest Energy (Kcal/mol)	No. of H-Bonds	H-Bonds Interaction Residues	Bond Distance (Å)
Eckol	−8.8	2	GLY34 SER36	3.277 3.239
Dieckol	−10.1	3	TRP76 THR232 LYS321	2.960 3.149 3.488
8,8′-Bieckol	−9.0	4	LYS107 GLY230 THR231 SER325	3.120 2.773 3.098 2.887

**Table 4 marinedrugs-17-00091-t004:** Molecular interactions of AChE with eckol, dieckol, and 8,8′-bieckol.

Compounds	Lowest Energy (Kcal/mol)	No. of H-Bonds	H-Bonds Interaction Residues	Bond Distance (Å)
Eckol	−8.8	5	THR83 TRP86 TYR124 SER125	2.855 2.712 3.134 2.883 & 3.313
Dieckol	−9.5	4	ASN233 THR238 ARG296 HIS405	3.399 2.837 3.344 3.181
8,8′-Bieckol	−9.2	1	ARG296	3.151

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
