# Peer review of "Dual BACE1 and Cholinesterase Inhibitory Effects of Phlorotannins from Ecklonia cava—An In Vitro and in Silico Study"

_marinedrugs, 2019, doi:10.3390/md17020091_

Round 1

Reviewer 1 Report

This is an interesting paper.

I suggest to improve the introduction with specific points

not all patients benefit of treatment explain possibly why

cite other review such as Marine Drug 2015 dec 25 14 (1) 5

There are different for of AChEdid the authors try to look at this?

Author Response

Dear Editor

We appreciate the thoughtful comments of the referees who provided critiques for this manuscript. These comments have provided us with a framework for the revision of the current manuscript. Uploaded are the revised manuscript and our responses to the reviewers’ comments and suggestions. In response to the reviewer comments, we have thoroughly revised manuscript as the referee requested, clarified several points and all changes were marked in red. With these changes, we believe that we have appropriately addressed all reviewer critiques in a clear and succinct fashion and that the revised manuscript has been significantly improved. We appreciate your reevaluation of the revised manuscript for publication in Marine Drugs. We hope this manuscript is now acceptable for publication.

Sincerely,

Mira Jun

Reviewer 1

I suggest to improve the introduction with specific points not all patients benefit of treatment explain possibly why cite other review such as Marine Drug 2015 dec 25 14 (1) 5. There are different of AChE did the authors try to look at this?

→ The authors included the difference of AChE in AD treatment in the Introduction part according to the reference the reviewer suggested as follows: In the past, it has been found that acetylcholinesterase (AChE) is involved in the degradation of the neurotransmitter. The observation of a significant loss of cholinergic neurons in AD patients is the major correlate of cognitive impairment. Cholinesterase inhibitors can increase acetylcholine levels in the synaptic cleft and partially ameliorate cognitive symptoms for patients with mild to severe AD (page 2, line 60-64).

Reviewer 2 Report

My recommendations you can see in the attachment.

Presented paper from Lee and Jun describes the effect of phlorotannins from edible brown seaweed as a potential treatment of Alzheimer´s Disease (AD).

I suggest a major revision of the manuscript.

Line 93: Please correct a mistake in the word sfor

Line 93: There is: IC50 were higher than resveratrol but according to table 1 were lower (their inhibitory effect was higher)

Line 98-102: The explanation of WHY the compounds were tested against other enzymes is missing. The connection between AD and the other enzymes should be mention in the introduction part.

Line 104: The chemical names should be added.

Line 110: In table 2, a more accurate description of the values is missing (e.g. units,…)

Line 169-175: This part of the discussion is more appropriate to the introduction

Line 216-247: This part predominantly consists of literally available information about the tested compound, there is no discussion of acquired results.

Line 225: Please add what is the source of AChE. Please delete BChE

Line 258: Please add more information about the method.

As this paper discuss the effect of polyphenolic compounds on AD, I strongly recommended supplementing two tests that are closely related to AD, namely inhibition of BChE and antioxidant properties test.

Author Response

Dear Editor

We appreciate the thoughtful comments of the referees who provided critiques for this manuscript. These comments have provided us with a framework for the revision of the current manuscript. Uploaded are the revised manuscript and our responses to the reviewers’ comments and suggestions. In response to the reviewer comments, we have thoroughly revised manuscript as the referee requested, clarified several points and all changes were marked in red. With these changes, we believe that we have appropriately addressed all reviewer critiques in a clear and succinct fashion and that the revised manuscript has been significantly improved. We appreciate your reevaluation of the revised manuscript for publication in Marine Drugs. We hope this manuscript is now acceptable for publication.

Sincerely,

Mira Jun

Reviewer 2

1) Line 93: Please correct a mistake in the word sfo

→ The authors corrected the typo on the text (page 3, line 110).

2) Line 93: There is: IC50 were higher than resveratrol but according to table 1 were lower (their inhibitory effect was higher

→ “Higher” was changed “lower” in the revised manuscript (page 3, line 110).

3) Line 98-102: The explanation of WHY the compounds were tested against other enzymes is missing. The connection between AD and the other enzymes should be mention in the introduction part.

→ In order to bring a BACE1 or AChE inhibitor to anti-AD, there are many challenges that must be overcome besides biological activity, molecular size and blood brain barrier penetration. Selectivity over other enzymes is one of the important challenges. Unlike BACE1, which leads to the formation of Aβ, α-secretase acts within Aβ domain to preclude Aβ generation in the non-amyloidogenic pathway. BACE1 and α-secretase compete for same APP substrate, with an increase in one cleavage event leading to a decrease in the other. Therefore, it is necessary to demonstrate that our tested compounds only inhibited BACE1 or AChE (page 2, line 54-57).

With serine proteases being found in nearly all body tissues and involved in various physiological function including digestion, reproduction and immune response, off target activity causing severe side effect is possible and likely if an inhibitor is not specific to BACE1 and AChE (Hedstrom, 2002). Therefore, to determine whether phlorotannins inhibited only targeted enzymes without affecting the normal pathway, all compounds were tested against TACE, candidate of α-secretases, and serine proteases. The above information was included in the Introduction of revised manuscript (page 3, line 118-123).

4) Line 104: The chemical names should be added.

→ The chemical names of phlorotannins were included as follows in revised manuscript: The chemical structures of eckol [4-(3,5-dihydroxyphenoxy)dibenzo-p-dioxin-1,3,6,8-tetrol], dieckol [4-[4-[6-(3,5-dihydroxyphenoxy)-4,7,9-trihydroxydibenzo-p-dioxin-2-yl]oxy-3,5-dihydroxyp henoxy]di

benzo-p-dioxin-1,3,6,8-tetrol], and 8,8′-bieckol [9-(3,5-dihydroxyphenoxy)-2-[9-(3,5-dihydroxypheno

xy)-1,3,6,8-tetrahydroxydibenzo-p-dioxin-2-yl]dibenzo-p-dioxin-1,3,6,8-tetrol] were shown in Figure 1 (page 3, line 103-107).

5) Line 110: In table 2, a more accurate description of the values is missing (e.g. units,…)

→ The accurate description of the values was added in Table 2: aThe inhibitory activity (%) were expressed as the mean±SD of three independent experiments. DMSO was used as negative control in TACE and serine proteases assays. bComparison of concentration level in each sample is not significantly different (page 4, line 139-141).

6) Line 169-175: This part of the discussion is more appropriate to the introduction

→ As reviewer suggested, the paragraph was moved to the introduction (page 2, line 84-90).

7) Line 216-247: This part predominantly consists of literally available information about the tested compound, there is no discussion of acquired results.

→ Please find the discussion of acquired results already mentioned in the front of Discussion part (page 9-10, line 189-252)

8) Line 225: Please add what is the source of AChE. Please delete BChE

→ The source of AChE was included. BchE was deleted (page 11, line 285).

9) Line 258: Please add more information about the method.

→ More information about the methods was included in Materials and methods section (page 11, line 290-318).

10) As this paper discuss the effect of polyphenolic compounds on AD, I strongly recommended supplementing two tests that are closely related to AD, namely inhibition of BChE and antioxidant properties test.

→Thanks for your valuable comments. As noted by reviewer, anti-BChE and antioxidant properties are closely related to AD. First, BChE is a serine hydrolase similar to AChE widely distributed throughout the central nervous system and catalyzes the hydrolysis of acetylcholine. Second, oxidative stress has been revealed to cause the formation of Aβ fibrils, which in turn accelerates oxidative stress, inflammatory responses and more Aβ accumulation, leading to ultimate cell death (Wilquet and Strooper, 2004; Jang et al., 2007). The inhibitory activity of our tested samples against BChE was evaluated in our preliminary study but did not show any significant inhibitory property against BChE, which was excluded in the present manuscript.

    Regarding antioxidant properties, three phlorotannins exhibited strong antioxidant properties against Aβ induced cytotoxicity using intracellular ROS scavenging activity in our previous study (Lee et al., 2019). Thus, at this point, the results of the evaluation of BACE1 and AChE inhibitory activity using in vitro and in silico analysis is sufficient to address the anti-AD properties of the tested compounds.

Reviewer 3 Report

The present is a good manuscript and adequated for Marine Drugs. The paper could be published in its present form

Author Response

Dear Editor

We appreciate the thoughtful comments of the referees who provided critiques for this manuscript. These comments have provided us with a framework for the revision of the current manuscript. Uploaded are the revised manuscript and our responses to the reviewers’ comments and suggestions. In response to the reviewer comments, we have thoroughly rev

ised manuscript as the referee requested, clarified several points and all changes were marked in red. With these changes, we believe that we have appropriately addressed all reviewer critiques in a clear and succinct fashion and that the revised manuscript has been significantly improved. We appreciate your reevaluation of the revised manuscript for publication in Marine Drugs. We hope this manuscript is now acceptable for publication.

Sincerely,

Mira Jun

Reviewer 4 Report

Comments to author

This manuscript studied the inhibitory effects of Phlorotannins from E. cava on BACE1 and cholinesterase. Overall, I find this manuscript interesting and well written. However, in my opinion, some of the results shown in the manuscripts need further control experiments or not well represented to have a better conclusion. I have some suggestions with respect to the presentation, incorporation of which will help the reader to understand in a better way My comments are listed below.

Major comments:

1. For Figure 2 and 3, In my opinion, there should be a panel for experiment done with known inhibitors (for example resveratrol for BACE1 (Figure 2), Galantamine for AchE (figure 3) as a positive control. The author can also make use of commercially available non-competitive inhibitor like L655,240 for the BACE1 experiment.

The author can also make use of negative control (which should not show inhibition with this enzyme in the study.  Also, it is not clear from the figure legends, how many time the experiment was repeated. The error bars are not clearly visible in the figure. Please mention that at the end of the figure legends.

2. BACE1 a member of the peptidase A1 family of aspartic proteases. In order to study the specificity towards BACE1, the author should also use an aspartic protease-like renin and cathepsin D in their study. Although they have used serine protease in their study for better comparison, an aspartic protease should be used.

Minor comments:

1. There are some spacing problems like in line no 93. Please correct that and check it throughout the manuscript.

2. Briefly explain the enzyme inhibition assays in the method section that will help the reader of the manuscript.

3. In table 1, the author mentions the log dose inhibition curve and data for it not shown anywhere. Is there any reason behind it? Also, mentioning the Km and Vmax in table 1 will help the reader in interpreting the data clearly.

Author Response

Dear Editor

We appreciate the thoughtful comments of the referees who provided critiques for this manuscript. These comments have provided us with a framework for the revision of the current manuscript. Uploaded are the revised manuscript and our responses to the reviewers’ comments and suggestions. In response to the reviewer comments, we have thoroughly revised manuscript as the referee requested, clarified several points and all changes were marked in red. With these changes, we believe that we have appropriately addressed all reviewer critiques in a clear and succinct fashion and that the revised manuscript has been significantly improved. We appreciate your reevaluation of the revised manuscript for publication in Marine Drugs. We hope this manuscript is now acceptable for publication.

Sincerely,

Mira Jun

Reviewer 4

Major comments:

1. For Figure 2 and 3, In my opinion, there should be a panel for experiment done with known inhibitors (for example resveratrol for BACE1 (Figure 2), Galantamine for AChE (figure 3) as a positive control. The author can also make use of commercially available non-competitive inhibitor like L655,240 for the BACE1 experiment.

→ The authors appreciate the reviewer’s valuable comments. It would be interesting to evaluate Dixon and Lineweaver-Burk plots with known inhibitors. However, the inhibition mode of known inhibitors such as resveratrol and L655,240 for BACE1 and galantamine for AChE has already been found in several studies (Greenblatt et al., 2004; Xu et al., 2008; Lee and Kang, 2010; Choi et al., 2011; Lu et al., 2012 ). The kinetic study in our present study aim to focus on inhibition mode of eckol, dieckol, and 8,8'-bieckol against target enzymes. Therefore, the biological study of our tested compounds is sufficient to account for the suitable BACE1 and AChE inhibitors.

2. The author can also make use of negative control (which should not show inhibition with this enzyme in the study.  Also, it is not clear from the figure legends, how many time the experiment was repeated. The error bars are not clearly visible in the figure. Please mention that at the end of the figure legends.

→ Thank you for your comments. As reviewer suggested, the figure legends were revised to be clear through adding a negative control and the number of times each experiment (line 130-131, 134-135, 139-141, 154, 159-160). In addition, in some graphs, like in Fig. 2 and 3, the error bars were not shown because the values were too small to be shown.

3. BACE1 a member of the peptidase A1 family of aspartic proteases. In order to study the specificity towards BACE1, the author should also use an aspartic protease-like renin and cathepsin D in their study. Although they have used serine protease in their study for better comparison, an aspartic protease should be used.

→ As the reviewer mentioned, since BACE1 is a member of the peptidase A1 family of aspartic proteases, the evaluating selectivity of phlorotannins toward other aspartic proteases will polish our manuscript. However, at this point, with serine proteases being found in nearly all body tissues and involved in various physiological function including digestion, reproduction and immune response, off target activity causing severe side effect is possible and likely if an inhibitor is not specific to BACE1 and AChE. Therefore, to determine whether phlorotannins inhibited only targeted enzymes without affecting the normal pathway, all compounds were tested against serine proteases in our study.

In addition, other previous studies, the results of only serine protease were enough to demonstrate the specificity of BACE1 (Park et al., 2004; Kwak et al., 2005; Jeon et., 2007; Song et al., 2008; Dai et al., 2010; Youn et al., 2012; Hu et al., 2016; Lee et al., 2018; Polito et al., 2018; etc). The above content was included in the revised manuscript (page 3, line 118-123) and the authors will design to evaluate the selectivity of samples toward other aspartic proteases like renin and cathepsin D in our future study.

Minor comments:

1. There are some spacing problems like in line no 93. Please correct that and check it throughout the manuscript.

→ The author corrected and checked spacing problems throughout the manuscript.

2. Briefly explain the enzyme inhibition assays in the method section that will help the reader of the manuscript.

→ The information of method was added in the revised manuscript in Materials and methods section (page 11, line 290-318).

3. In table 1, the author mentions the log dose inhibition curve and data for it not shown anywhere. Is there any reason behind it? Also, mentioning the Km and Vmax in table 1 will help the reader in interpreting the data clearly.

→ Thanks for your valuable comments. The log dose inhibition curve was added in revised manuscript (page 4, line 129-131).

In vitro kinetic and inhibition studies are useful tools for predicting in vivo pharmacokinetics and the potential for drug-drug interactions. A non-competitive inhibitor usually binds somewhere other than the active site, but is able to change the conformation of the active site in such a way that the substrate is not able to efficiently catalyze the reaction. The inhibition level is dependent on the concentration of the inhibitor but is not reduced by increasing concentrations of substrate. Because of this, Vmax is reduced, but Km is unaffected. In BACE1 inhibitory activity, phlorotannins decreased the Vmax values without affecting the affinity of BACE1 toward the Km which demonstrated that these compounds exhibited non-competitive inhibition against BACE1. However, in AChE inhibitory study, our compounds displayed competitive type of inhibition, with unchanged Vmax and increased Km. The above paragraph was added in revised manuscript (page 9-10, line 227-236).

Round 2

Reviewer 2 Report

Accept in present form

Reviewer 4 Report

I am satisfied with the response from the authors and I think the manuscript is ready to be accepted for publication.